# Extracting representations of cognition across neuroimaging studies improves brain decoding

**Arthur Mensch** [1]*, **Julien Mairal** [2], **Bertrand Thirion** [1], **Gaël Varoquaux** [1]

**1** Inria, CEA, Univ. Paris Saclay, Palaiseau, France, **2** Univ. Grenoble Alpes, Inria, CNRS, Grenoble INP, LJK, Grenoble, France

* arthur.mensch@m4x.org

**Data Availability Statement:** All data files and resulting atlases are available on the website cogspaces.github.io.

**Funding:** This project has received funding from the European Union's Horizon 2020 Framework

## Abstract

Cognitive brain imaging is accumulating datasets about the neural substrate of many different mental processes. Yet, most studies are based on few subjects and have low statistical power. Analyzing data across studies could bring more statistical power; yet the current brain-imaging analytic framework cannot be used at scale as it requires casting all cognitive tasks in a unified theoretical framework. We introduce a new methodology to analyze brain responses across tasks without a joint model of the psychological processes. The method boosts statistical power in small studies with specific cognitive focus by analyzing them jointly with large studies that probe less focal mental processes. Our approach improves decoding performance for 80% of 35 widely-different functional-imaging studies. It finds commonalities across tasks in a data-driven way, via common brain representations that predict mental processes. These are brain networks tuned to psychological manipulations. They outline interpretable and plausible brain structures. The extracted networks have been made available; they can be readily reused in new neuro-imaging studies. We provide a multi-study decoding tool to adapt to new data.

## Author summary

Brain-imaging findings in cognitive neuroscience often have low statistical power, despite the availability of functional imaging data across hundreds of studies. Yet, with current analytic frameworks, combining data across studies that map responses to different tasks discards the nuances of the cognitive questions they ask. In this paper, we propose a new approach for fMRI analysis, where a predictive model is used to extract the shared information from many studies together, while respecting their original paradigms. Our method extracts cognitive representations that associate a wide variety of functions to specific brain structures. This provides quantitative improvements and cognitive insights when analyzing together 35 task-fMRI studies; the breadth of the functional data we consider is much higher than in previous work. Reusing the representations learned by our approach also improves statistical power in studies outside the training corpus.

Programme for Research and Innovation under grant agreement N785907 (Human Brain Project SGA2). Arthur Mensch was supported by a grant from the Labex DigiCosme (AMPHI project). Julien Mairal was supported by the ERC grant SOLARIS (N714381) and a grant from ANR (MACARON project ANR-14-CE23-0003-01). The funders had no role in study design, data collection and analysis, decision to publish, or preparation of the manuscript.

**Competing interests:** The authors have declared that no competing interests exist.

## Introduction

Cognitive neuroscience uses functional brain imaging to probe the brain structures underlying mental processes. The field is accumulating neural activity responses to specific psychological manipulations. The diversity of studies that probe different mental processes gives a big picture on cognition [1]. However, as brain mapping has progressed in exploring finer aspects of mental processes, the statistical power of studies has stagnated or even decreased [2]—although sample size is increasing over years, it has not kept pace with the reduction of effect size. As a result, many, if not most individual studies often have low statistical power [3]. Large-scale efforts address this issue by collecting data from many subjects [4, 5]. For practical reasons, these efforts however focus on a small number of cognitive tasks. In contrast, establishing a complete view of the links between brain structures and the mental processes that they implement requires varied cognitive tasks [6], each crafted to recruit different mental processes. In this paper, we develop an analysis methodology that pools data across many task-fMRI studies to increase both statistical power and cognitive coverage. Standard meta analyses can only address *commonalities* across studies, as they require casting mental manipulations in a consistent overarching cognitive theory. They can bring statistical power at the cost of coverage and specificity in the cognitive processes. On the opposite, our approach uses the *specific* psychological manipulations of each study and extracts shared information from the brain responses across paradigms. As a result, it improves markedly the statistical power of mapping brain structures to mental processes. We demonstrate these benefits on 35 functional-imaging studies, all analyzed accordingly to their individual experimental paradigm.

Interpreting overlapping brain responses calls for multivariate analyses such as brain decoding [7]. Brain decoding uses machine learning to predict mental processes from the observed brain activity. It is a crucial tool to associate functions to given brain structures. Such inference endeavor calls for decoding across cognitive paradigms [8]. Indeed, a single study does not provide enough psychological manipulations to characterize well the functions of the brain structures that it activates [6], while covering a broader set of cognitive paradigms gives more precise functional descriptions. Moreover, the statistical power of functional data is limited by the sample size [3]. A single study seldom provides more than few hundreds of observations, which is well below machine-learning standards. Open repositories of brain functional images [9, 10] bring the hope of large-scale decoding with much larger sample sizes.

Yet, shoehorning such a diversity of studies into a decoding problem requires daunting manual annotation to build explicit correspondences across cognitive paradigms. We propose a different approach: we treat the decoding of each study as a single task in a multi-task linear decoding model [11, 12]. The parameters of this model are partially shared across studies to enable discovering potential commonalities. Model fitting—the training step of machine learning—is performed jointly, using non-convex training and regularization techniques [13, 14]. We thus learn to perform simultaneous decoding in many studies, to leverage the brain structures that they implicitly share. The extracted structures provide universal priors of functional mapping that improve decoding on new studies and can readily be reused in subsequent analyzes.

Models that generalize in measurable ways to new cognitive paradigms would ground broader pictures of cognition [15]. However, they face the fundamental roadblock that each cognitive study frames a particular question and resorts to specific task oppositions without clear counterpart in other studies [16]. In particular, a cognitive fMRI study results in *contrast* brain maps, each of which corresponds to an elementary psychological manipulation, often unique to a given protocol. Analyzing contrast maps across studies requires to model the relationships between protocols, which is a challenging problem. It has been tackled by labeling

common aspects of psychological manipulations across studies, to build decoders that describe aspects of unseen paradigms [17, 18]. This annotation strategy is however difficult to scale up to a large set of studies as it requires expert knowledge on each study. The lack of complete cognitive ontologies to decompose psychological manipulations into mental processes [19] makes it even harder.

To overcome these obstacles, our multi-study decoding approach relies on the *original* labels of each study. Instead of relabeling data into a common ontology, the method extracts data-driven common cognitive dimensions. Our guiding hypothesis is that activation maps may be accurately decomposed into latent components that form the neural building blocks underlying cognitive processes [20]. This modelling overcomes the limitations of single-study cognitive subtraction models [19]. In particular, we show that it improves statistical power in individual studies: it gives better decoding performance for a vast majority of studies, and the improvement is particularly pronounced for studies with a small number of subjects. Our implicit modelling of functional information has the further advantage of providing explainable predictions. It decomposes the common aspects of psychological manipulations across studies onto latent factors, supported by spatial brain networks that are *interpretable* for neuroscience. These form by themselves a valuable resource for brain mapping: a functional atlas tuned to jointly decoding the cognitive information conveyed by various protocols. The trained model is a deep *linear* model. Building a linear model is important to bridge with classic decoding techniques in neuroimaging and ensures interpretability of intermediary representations.

## Materials and methods

We first give an informal overview of the contributed methods for multi-study decoding. We review the mathematical foundations of the methods in a second part—a complete description is provided in S1 Appendix. Finally, we describe how we validate the performance and usability of the approach. A preliminary version of our method was described in [21], with important differences and a less involved validation (discussed in details in S1 Appendix).

### Method overview

The approach has three main components, summarized in Fig 1: aggregating many fMRI studies, training a deep linear model, and reducing this model to extract *interpretable* intermediate representations. These representations are readily reusable to apply the methodology to new data. Building upon the increasing availability of public task-fMRI data, we gathered statistical maps from many task studies, along with rest-fMRI data from large repositories, to serve as training data for our predictive model (Fig 1A). Statistical maps are obtained by standard analysis, computing z-statistics maps for either base conditions, or for contrasts of interest when available. We use 40,000 subject-level contrast maps from 35 different studies (detailed in Table 1), with 545 different contrasts; a few are acquired in cohorts of hundreds of subjects (e.g., HCP, CamCan, LA5C), but most of them feature more common sample sizes of 10 to 20 subjects. These studies use different experimental paradigms, though most recruit related aspects of cognition (e.g., motor, attention, judgement tasks, object recognition).

We use machine-learning classification techniques for inter-subject decoding. Namely, we associate each brain activity contrast map with a predicted contrast class, chosen among the contrasts of the map's study. For this, we propose a linear classification model featuring *three* layers of transformation (Fig 1B). This architecture reflects our working hypothesis: cognition can be represented on basic functions distributed spatially in the brain. The first layer projects

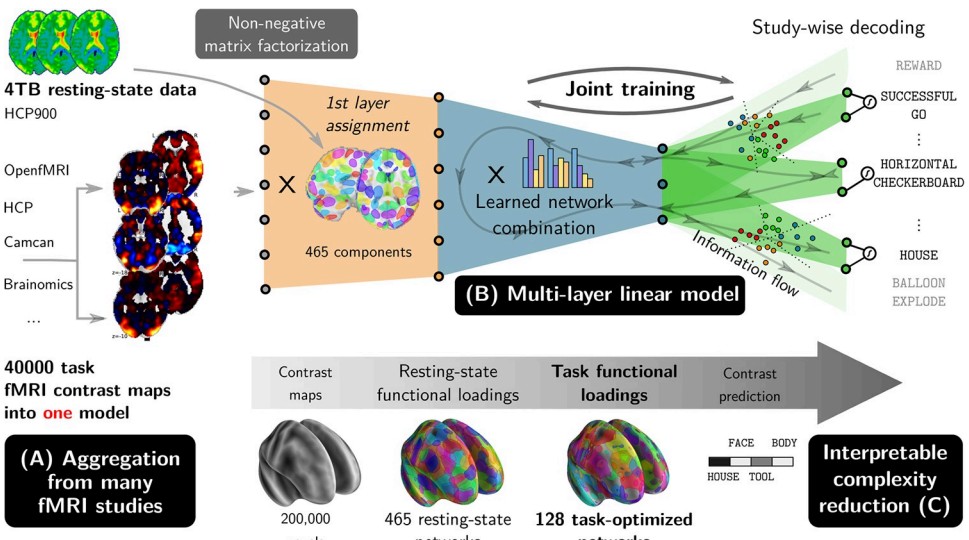

**Fig 1. General description of our multi-study decoding approach.** We perform inter-subject decoding using a shared three-layer model trained on multiple studies. An initial layer projects the input images from all studies onto functional networks learned on resting-state data. Then, a second layer combines the functional networks loadings into common meaningful cognitive subspaces that are used to perform decoding for each study in a third layer. The second and third layers are trained jointly, fostering transfer learning across studies.

contrast maps onto $k = 465$ functional units learned from resting-state data. This first dimension reduction should be interpreted as a projection of the brain signal onto small, smooth and connected brain regions, tuned to capture the resting-state brain signal with a fine grain. The second layer performs dimension reduction and outputs an embedding of the brain activity into $l = 128$ features that are *common* across studies. The embedded data from each study are then classified into their respective contrast class using a study-specific classification output from the third layer, in a setting akin to multi-task learning (see [55] for a review).

The second layer and the third layer are jointly extracted from the task-fMRI data using regularized stochastic optimization. Namely, the shared brain representation is optimized simultaneously with the third layer that performs decoding for every study. In particular, we use dropout regularization [56] in the layered model and stochastic optimization [13] to obtain good out-of-sample performance.

Study-specific decoding is thus performed on a shared low-dimensional brain representation. This representation is supported on 128 different combinations of the first 465 functional units identified with resting-state data. These combinations form diffuse networks of brain regions, that we call *multi-study task-optimized networks* (MSTONs). MSTONs differ from the notion of brain networks in the neuroscience literature—the later are typically obtained using a low-rank factorization of resting-state data, with a much lower number of components ($k \approx$ 20) than what we use to extract the *functional units* of the first layer.

As we will show, projecting data onto MSTONs improves across-subject predictive accuracy, removing confounds while preserving the cognitive signal. Interpretability is guaranteed by the linearity of the model and a post-training identification of stable directions in the space of latent representations. These networks capture a general multi-study representation of the cognitive signal contained in statistical maps.

**Table 1. Training and experiment set of fMRI studies.** Note that even though some tasks are similar, they may feature different contrasts. Task correspondence is not encoded explicitly in our model. Table C in S1 Appendix lists each contrast used in each study.

| Study and task description | # contrasts | # subjects |
|---|---|---|
| [22] High level math & Localizer | 31 | 30 |
| [23] The ARCHI project | 30 | 78 |
| [24] Brainomics | 19 | 94 |
| [25] CamCAN | 5 | 605 |
| [26, 27] Music structure & Sentence structure | 19 | 35 |
| [28] Sentence/music complexity | 25 | 20 |
| [29] Balloon Analog Risk-taking | 12 | 16 |
| [30] Baseline trials & Classication learning | 7 | 17 |
| [31] Rhyme judgment | 3 | 13 |
| [32] Mixed-gambles | 4 | 16 |
| [33] Plain or mirror-reversed text | 9 | 14 |
| [34] Stop-signal | 6 | 20 |
| [35] Conditional stop-signal & Stop-signal | 12 | 13 |
| [36] Balloon analog risk task & Emotion regulation & Stop-signal & Temporal discounting task | 23 | 24 |
| [37] Classification probe without feedback & Dual-task weather classification & Single-task weather classification & Tone-counting | 14 | 14 |
| [38] Classification learning & Stop-signal | 11 | 8 |
| [38] Classification learning & Stop-signal | 11 | 8 |
| [39] Cross-language repetition priming | 17 | 13 |
| [40] Classification learning | 3 | 13 |
| [41] Simon task | 8 | 7 |
| [7] Visual object recognition | 13 | 6 |
| [42] Word & object processing | 6 | 49 |
| [43] Emotion regulation | 26 | 34 |
| [44] False belief | 7 | 36 |
| [45] Incidental encoding | 26 | 18 |
| [46] Covert verb generation & Line bisection & Motor & Overt verb generation & Overt word repetition | 11 | 10 |
| [47] Auditory oddball & Visual oddball | 8 | 17 |
| [48] Continuous house vs face & Discontinuous house (800ms) vs face & Discoutinuous house (400ms) vs face & House vs face | 30 | 11 |
| [48] Continuous house vs face & House vs face | 23 | 13 |
| [49] The Human Connectome Project | 23 | 786 |
| [50] Face recognition | 5 | 16 |
| [51] Arithmetic & Saccades | 26 | 19 |
| [52] UCLA LA5C consortium | 24 | 189 |
| [53] Foreign language & Localizer & Saccade | 34 | 65 |
| [54] Auditory compression & Visual compression | 14 | 16 |
| Total | 545 | 2343 |

## Mathematical modelling

Following this informal description, we now review the mathematical foundations of our decoding approach. The complete descriptions of the predictive models and of the training algorithms are provided in S1 Appendix.

We consider $N$ task-fMRI studies, that we use for functional decoding. In this setting, each study $j$ features $n^j$ subjects, for which we compute $c^j$ different contrasts maps, using the General Linear Model [57]. Masking them using a grey-mask filter in the MNI space, we obtain a set of $z$-maps $(\boldsymbol{x}_j^i)_{j \in [1, c^j n^j]}$, in $\mathbb{R}^p$, that summarizes the effect on brain activations of the psychological conditions $(y_i^j)_{i \in [1, c^j n^j]}$. The goal of functional decoding is to learn a predictor from $z$-maps to psychological conditions, namely a function $f^j : \mathbb{R}^p \rightarrow [1, c^j]$. This predictor will be evaluated on unseen subjects for validation.

**Linear decoding with shared parameters.** In our setting, we couple the predictors $(f^j)_{j \in [N]}$ by forcing them to share parameters. Each study corresponds to a classification task, and we cast the problem as multi-task learning (as first considered in [11]). For this, we consider a given $z$-map $\boldsymbol{x}_i^j$ in study $j$. We compute the predicted psychological condition using a factorized linear model:

$$\hat{y}_i^j = f^j(\boldsymbol{x}_i^j) = \underset{k \in [1, c^j]}{\operatorname{argmax}} \, (\boldsymbol{U}^j \boldsymbol{L} \boldsymbol{D} \boldsymbol{x}_i^j + \boldsymbol{b}^j)_k.$$

The matrix $\boldsymbol{D} \in \mathbb{R}^{k \times p}$ and $\boldsymbol{L} \in \mathbb{R}^{l \times k}$ contain the basis for performing two successive projection of the $z$-map $\boldsymbol{x}_j^i$ onto low-dimension spaces. Those parameters are shared over all studies $j \in [N]$ and form the first and second layer of our model. The matrix $\boldsymbol{U}^j \in \mathbb{R}^{l \times c^j}$ and the bias vector $\boldsymbol{b}^j \in \mathbb{R}^{c^j}$ are the parameters of a multi-class linear classification model that labels the projected map $\boldsymbol{L} \boldsymbol{D} \boldsymbol{x}_i^j$ with a psychological condition within the study $j$. Those parameters are specific to each study $j$, and form the third layer of our model.

**First layer training from resting-state data.** The first dimension reduction, contained in the matrix $\boldsymbol{D} \in \mathbb{R}^{k \times p}$, is learned using external resting-state data, from the HCP project [4]. Voxel time-series are stacked in a data matrix $\boldsymbol{X} \in \mathbb{R}^{n \times p}$ (with 4 millions brain-images), that is factorized so that $\boldsymbol{X} \approx \boldsymbol{D} \boldsymbol{A}$, with $\boldsymbol{D}$ non-negative and sparse (i.e. with mostly null coefficients). This forces the elements of $\boldsymbol{D}$ to delineate localized functional units. We use a sparse non-negative matrix factorization objective [58] and a recent scalable matrix factorization algorithm [59] to learn $\boldsymbol{D}$, as detailed in S1 Appendix. The non-negativity constraint allows to interpret functional units as a soft parcellation of the brain. We do not use additional spatial constraints, as non-negative sparse matrix factorization with $k = 465$ components readily finds smooth connected regions.

**Joint training of the second and third layer.** The matrix $\boldsymbol{L}$ and the multiple matrices $(\boldsymbol{U}^j)_{j \in [n]}$ and intercepts $(\boldsymbol{b}^j)_j$ are trained jointly to minimize the objective

$$\min_{\boldsymbol{L}, \{\boldsymbol{U}^j, j \in [N]\}} \sum_{j=1}^{N} \frac{1}{n^j} \sum_{i=1}^{c^j n^j} \ell_j(\boldsymbol{U}^j \boldsymbol{L} \boldsymbol{D} \boldsymbol{x}_i^j + \boldsymbol{b}^j, y_i^j),$$

where $\ell_j$ is the standard $\ell_2$-regularized multinomial loss function for training a linear model with $c^j$ classes (see S1 Appendix for details). This objective is trained using Adam [13]; at each step, we select a batch of examples from one study. To prevent specialization of the rows of matrix $\boldsymbol{L}$ to specific studies, we add a dropout noise [14] to the activations $\boldsymbol{D} \boldsymbol{x}_i^j$ and $\boldsymbol{L} \boldsymbol{D} \boldsymbol{x}_i^j$ during training.

**Model consensus.** Although the atoms of $\boldsymbol{D}$ are naturally interpretable, the fact that the product $\boldsymbol{U}^j \boldsymbol{L}$ can always be rewritten as $\boldsymbol{U}^j \boldsymbol{M}^{-1} \boldsymbol{M} \boldsymbol{L}$ for an invertible matrix $\boldsymbol{M}$ prevents us from directly identifying meaningful directions in the low-dimensional space spanned by $\boldsymbol{L} \boldsymbol{D}$. On the other hand, we found this space to be remarkably stable across training runs. We therefore propose an ensemble technique to extract a non-negative matrix $\bar{\boldsymbol{L}} \in \mathbb{R}^{l \times k}$ such that $\bar{\boldsymbol{L}} \boldsymbol{D}$

captures meaningful directions (as above-mentioned non-negativity enables us to interpret MSTONs as soft brain parcellations).

For this, we train $R$ decoding models with different sampling order and initialization, to obtain $(L_r)_{r \in [R]}$. We stack these matrices into a tall matrix $\tilde{L} \in \mathbb{R}^{l\,R \times k}$, that we factorize as $\tilde{L} = K\bar{L}$, with $\bar{L} \in \mathbb{R}^{l \times k}$ non-negative and sparse. This is in turn (see S1 Appendix) yields a consensus model $(D, \bar{L}, (\bar{U}^j, \bar{b}^j)_{j \in [N]})$, where $\bar{L}D \in \mathbb{R}^{l \times p}$ is sparse and non-negative. It therefore holds interpretable brain networks, learned in a supervised manner from many studies—those form the MSTONs.

**Layer widths.** We chose $k = 465$ and $l = 128$ as those are a good compromise between model performance and interpretability—trade-offs in choosing the number of functional units $k$ for fMRI analysis are discussed in e.g. [60], and we compare the model performance for different $l$ in Fig E in S1 Appendix. Choosing $l$ smaller than the number of classes enforces a low-rank structure over the set of 545 classification maps.

## Validation

**Quantitative measurements.** The benefits of multi-study decoding may vary from study to study, and a single number cannot properly quantify the impact of our approach. We measure decoding *accuracy* on left-out subjects (half-split, repeated 20 times) for each study. For each split and each study, we compare the scores obtained by our model to results obtained by simpler baseline decoders, that classify contrast maps separately for each study, and directly from voxels. To analyse the impact of our method on the prediction accuracy specifically for each contrast, we also report the *balanced-accuracy* for each predicted class. For completeness, we report mean accuracy gain and the number of studies for which multi-study decoding improves accuracy—those hint at the benefit that one may expect when applying the method to a new fMRI study. Mathematical definitions of the metrics in use are reported in S1 Appendix, Section C.2.

**Exploring MSTONs.** Our model optimizes its second and third layers to project brain images on representations that help decoding. These representations boil down to MSTONs combinations: MSTONs form a valuable output of the model, as they can easily be reused to project data for new decoding tasks. We provide 2D and 3D views of the MSTONs, showing how they cover the brain. We evaluate the importance of each network for decoding a certain contrast by computing the cosine similarity between the MSTON and the classification map associated with this contrast. We represent these contrasts' names as specified in their original studies with word-clouds, with a size increasing with their similarity with a given MSTON.

**Classification maps.** As our model is linear, we qualitatively compare the classification maps that it yields with maps obtained with a baseline single-study voxel-level decoding approach. For both approaches, we compute the correlation matrix between classification maps to uncover potential clusters of similar maps, using hierarchical clustering [61]. We compare this correlation matrix in term of how clustered it is, using the cophenetic correlation coefficient [62] and the mean absolute cosine similarity between maps.

## Reusable tools and resources

Our approach can be used to improve statistical power of decoding in new fMRI studies. To facilitate its use, we have released resources and the *cogspaces* library (http://cogspaces.github.io). We include software to train the models. Pre-trained MSTONs networks (with associated word-clouds) can be downloaded and inspected on a dedicated page (https://cogspaces.github.io/assets/MSTON/components.html). The statistical maps used in the present study may be

downloaded using our library, or on neurovault.org. The published MSTON networks hold the representations extracted from the 35 studies that we have considered.

## Results

We first detail the quantitative improvements brought by our approach, before exploring these results from a cognitive neuroscience point of view.

### Improved statistical performance of multi-study decoding

Decoding from multi-study task optimized networks gives quantitative improvements in prediction of mental processes, as summarized in Fig 2. For 28 out of the 35 task-fMRI studies that we consider, the MSTON-based decoder outperforms single-study decoders (Fig 2A). It improves accuracy by 17% for the top studies, with a mean gain of 5.8% (80% experiments with net increase, 4.8% median gain) across studies and cross-validation splits (Fig 2B). *Jointly* minimizing errors on every study constructs second-layer representations that are efficient for many study-specific decoding tasks; the second layer parameters therefore incorporate information from all studies. This shared representation enables information transfer among the many decoding tasks performed by the third layer—predictive accuracy is thus improved thanks to *transfer learning*. Although we have not explicitly modeled how mental processes or psychological manipulations are related across experiments, our quantitative results show that these relations can be captured by the model—encoded into the second layer—to improve decoding performance.

Studies with diverse cognitive focus benefit from using multi-study modeling. The different decoding tasks have varying difficulties—we report performance sorted by chance level in Fig L in S1 Appendix. Among the highest accuracy gains, we find cognitive control (stop-signal), classification studies, and localizer-like protocols. Our corpus contains many of such studies;

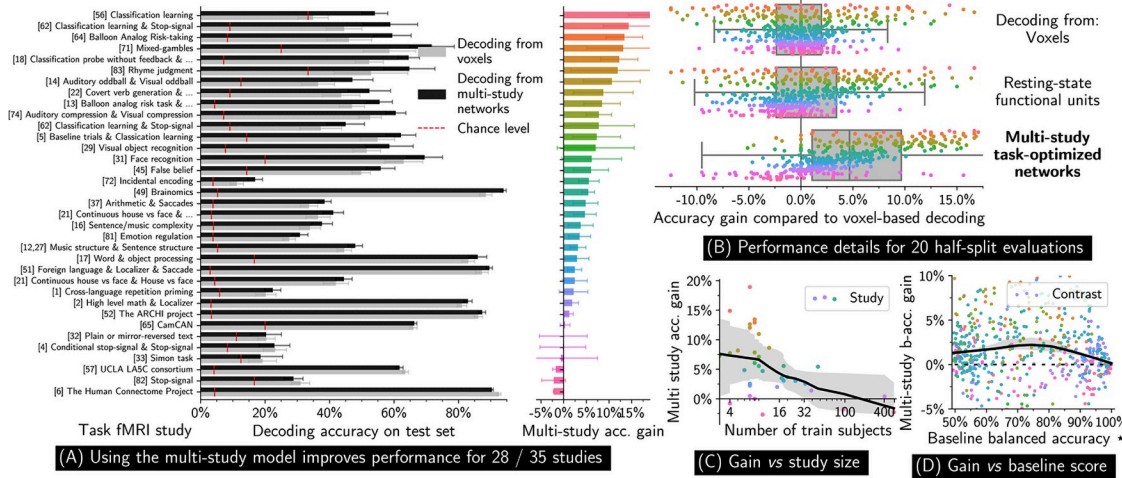

**Fig 2. Quantitative performance of multi-study decoding. (A)** Multi-study decoding improves the performance of cognitive task prediction across subjects for most studies. **(B)** Overall, decoding from task-optimized networks leads to a mean improvement accuracy of 5.8% compared to voxel or networks based approaches. Each point corresponds to a study and a train/test split. **(C)** Studies of typical size strongly benefit from transfer learning, whereas little information is gained for very large studies. **(D)** Contrasts that are moderately difficult to decode benefit most from transfer. Error bars are calculated over 20 random data half-split. *(D) shows *per-contrast* balanced accuracy (50% chance level), whereas *per-study* classification accuracy is used everywhere else. Numbers are reported in Table A in S1 Appendix.

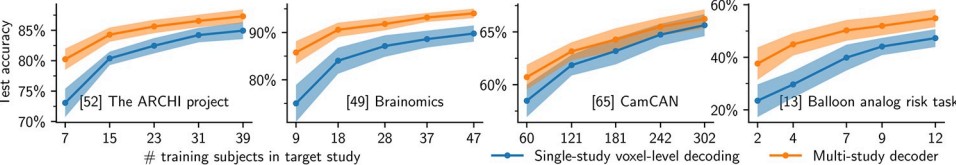

**Fig 3. Varying accuracy improvement with study size.** Training an MSTON decoder increases decoding accuracy for many studies (see Fig 2A). Gains are higher as we reduce the number of training subjects in target studies—pooling multiple studies is especially useful to decode studies performed on small cohorts. Error bars are calculated over 20 random data half-splits.

as a result, multi-study decoding has access to many more samples to gather information on the associated cognitive networks. The activation of these networks is better captured in the shared part of the model, thereby leading to the observed improvement. In contrast, for a few studies, among which HCP and LA5C, we observe a slight negative transfer effect. This is not surprising—as HCP holds 900 subjects, it may not benefit from the aggregation of much smaller studies; LA5C focuses on higher-level cognitive processes with limited counterparts in the other studies, which precludes effective transfer.

Fig 2B shows that simply projecting data onto resting state functional networks instead of using our three-layer model does not significantly improve decoding, although the net accuracy gain varies from study to study. Adding a second *task-optimized*—supervised—dimension reduction is thus necessary to improve overall decoding accuracy. Functional contrasts that are either easy or very hard to decode do not benefit much from multi-study modeling, whereas classes with a balanced-accuracy around 80% experience the largest decoding improvement (Fig 2). We attribute this to two causes: easy-to-decode studies do not benefit from the extra signal provided by other studies, while some studies in our corpus are simply too hard to decode due to a low signal-to-noise ratio. Fig 2D shows that the benefit of multi-study modeling is higher for smaller studies, confirming that the proposed method boosts their inter-subject decoding performance.

In Fig 3, we vary the number of training subjects in target studies, and compare the performance of the multi-study decoder with a more standard one. We observe that the smaller the study size, the larger the performance gain brought by multi-study modeling. Transfer learning in inter-subject decoding is thus particularly effective for small studies (e.g., 16 subjects), that still constitute the essential of task-fMRI studies. To confirm this effect, we trained a multi-study model on a subset of 15 subjects per study, considering studies that comprise more than 30 subjects. In this case, the transfer learning effect is positive for all studies (Fig K in S1 Appendix), including those for which negative transfer was observed when using full cohorts.

Finally, we show in Fig B in S1 Appendix that training a three-layer model and reusing the first two layers as a fixed dimension reduction when decoding a new study improves decoding accuracy on average. The extracted functional networks (MSTONs) thus provide a study-independent prior that is likely to improve decoding for studies probing different cognitive questions than the ones considered in the training corpus.

## Multi-study task-optimized networks capture broad cognitive domains

We outline the contours of the 128 extracted MSTONs in Fig 4A. The networks almost cover the entire cortex, a consequence of the broad coverage of cognition of the studies we gathered. Task-optimized networks must indeed capture information to predict 545 different cognitive

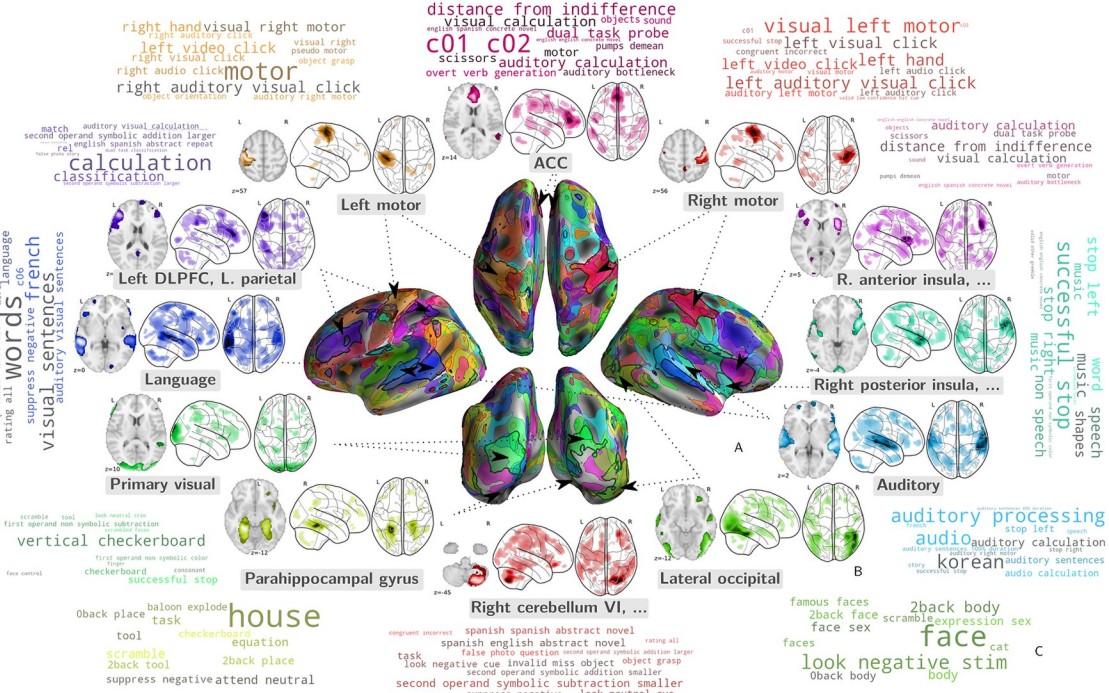

**Fig 4. Visualization of some of task-optimized networks.** Our approach learns networks that are important for decoding across studies. These networks are individually focal and collectively well spread across the cortex. They are readily associated with the cognitive tasks that they contribute to predict. We display a selection of these networks on the cortical surface **(A)** and in 2D transparency **(B)**, named with the salient anatomical brain region they recruit, along with a word-cloud **(C)** representation of the stimuli whose likelihood increases with the network activation. The words in this word cloud are the terms used in the contrast names by the investigators; they are best interpreted in the context of the corresponding studies.

classes from the resulting distributed brain activity. Brain regions that are systematically recruited in task-fMRI protocols, e.g., motor cortex, auditory cortex, and primary visual cortex, are finely segmented by MSTON: they appear in several different networks. Capturing information in these regions is crucial for decoding many contrasts in our corpus, hence the model dedicates a large part of its representation capability to it. As decoding requires capturing distributed activations, MSTON are formed of multiple regions (Fig 4B). For instance, both parahippocampal gyri appear together in the yellow bottom-left network.

Most importantly, Fig 4B and 4C show that the model relates extracted MSTONs to specific cognitive information. The MSTONs each play a role in decoding a subset of contrasts. Components may capture low-level or high-level cognitive signal, though the low-level components are easier to interpret. Indeed, at a lower level, they outline the primary visual cortex, associated with contrasts such as checkerboard stimuli, and both hand motor cortices, associated with various tasks demanding motor functions. At a higher level, some interpretable components single out the left DLPFC and the IPS in separate networks, used to decode tasks related to calculation and comparison. Others delineate the language network and the right posterior insula, important in decoding tasks involving music [27]. Yet another MSTON delineates Broca's area, associated with language tasks (Fig 5).

Inspecting the tasks associated with the MSTONs reveals structure-function links. Once again, the results are more interpretable for low-level functions, although some well-known high-level functional associations are also well captured. For instance, several components on

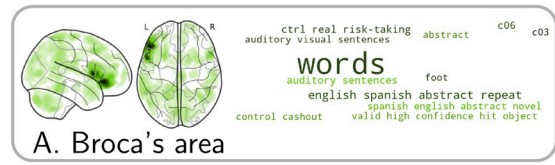

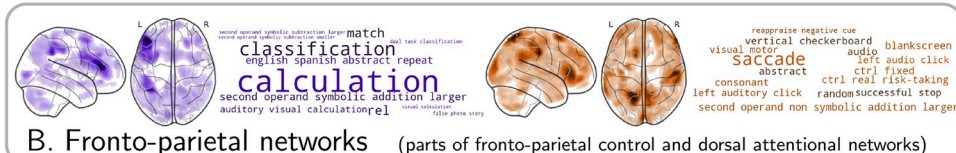

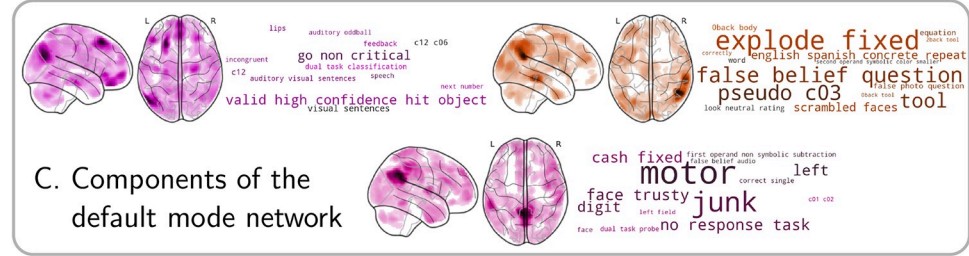

**Fig 5. Task-optimized networks associated with high-level functions.** Some MSTONs outline brain-circuits that are associated with language, e.g. Broca's area (**A**), or more abstract functions, e.g. fronto-parietal networks (**B**) or even part of the default mode network (**C**). Those networks are more distributed than the ones displayed in Fig 4, but are associated with relatively interpretable word-clouds.

Fig 4 involve brain regions recruited across a wide variety of tasks, such as the anterior insula, engaged in auditory and visual tasks [63] and considered to tackle ambiguous perceptual information, or the ACC, associated with tasks with affective components [64] and reward-based decision making [65]. Some MSTONs are more distributed, but correspond to well-known patterns brain activity. For example, Fig 5 show components that reveal parts the default mode networks –associated with baseline conditions, theory-of-mind tasks and prospection [66, 67]–, parts of the fronto-parietal control network –associated with a variety of problem-solving tasks [68]– and the dorsal attentional network –associated with visuo-spatial attention tasks such as saccades [69].

Visualizing MSTONs along with word-clouds serves essentially an illustratory purpose. It yields more interpretable results with focal networks than with distributed networks. In both cases, the words in the contrasts related to the given MSTONs capture documented structure-function associations. Interpretability may be improved by reducing the number of extracted networks, at the cost of a quantitative loss in performance. In particular, with $k = 128$ components, the default mode network is split across several MSTONs (Fig 5). Such a splitting is common for high-dimensional decomposition of the fMRI signal, as noted in resting state [70], as a network such as the default-mode network has different sub-units with distinct functional contributions [71]. Conversely, some contrast maps are correlated with several distributed MSTONs, as illustrated in Fig A in S1 Appendix.

## Impact of multi-study modeling on classification maps

To better understand how multi-study training and layered representations improve decoding performance, we compare classification maps obtained using our model to standard decoder

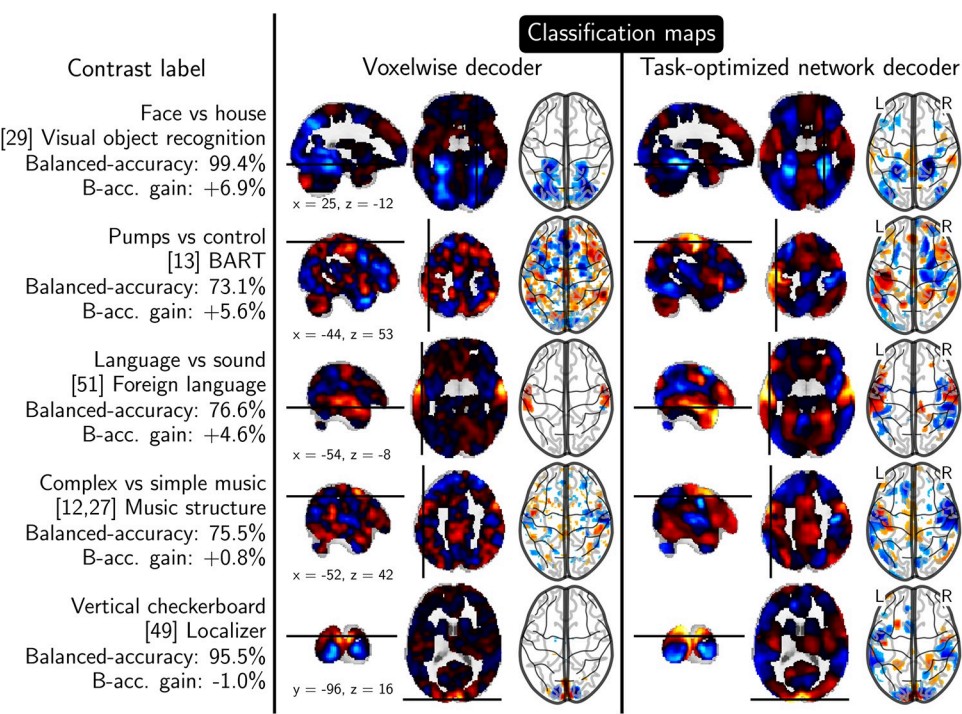

**Fig 6. Classification maps obtained from multi-study decoding** (right). The maps are smoother and more focused on functional modules than when decoding from voxels (left). For contrasts for which there is a performance boost (top of the figure), relevant brain regions are better delineated, as clearly visible on the face vs house visual-recognition opposition, in which the fusiform gyrus stands out better. B-acc stands for balanced accuracy using multi-study decoding (see text).

maps in Fig 6. For contrasts with significant accuracy gains, the classification maps are less noisy and more focal. They single out determinant regions more clearly, e.g., the fusiform face area (FFA, row 1) in classification maps for the face-vs-house contrast, or the left motor cortex in maps (row 2) predicting pumping action in BART tasks [29]. The language network is typically better delineated by our model (row 3), and so is the posterior insula in music-related contrasts (row 4). These improvements are due to two aspects: First, projecting onto a lower dimension subspace has a denoising effect on contrast maps, that is already at play when projecting onto simple resting-state functional networks. Second, multi-study training finds more scattered classification maps, as these combine complex MSTONs, learned on a large set of brain images. Our method slightly decreases performance for a small fraction of contrasts, such as maps associated with vertical checkerboard (row 5), a condition well localized and easy to decode from the original data. Our model renders them too much distributed, an unfortunate consequence of multi-study modeling.

We also compare original input contrast maps to their transformation by the projection on task-optimized networks (Fig C in S1 Appendix). Projected data are more focal, i.e. spatial variations that are unlikely to be related to cognition are smoothed. This offers a new angle on the quantitative results (Fig 2): brain activity expressed as the activation of these networks captures better cognition and allows decoders to generalize better across subjects than when classifying raw input directly.

**Information transfer among classification maps.** In Fig 7, we compare the correlation between the 545 classification maps obtained using a multi-study decoder and using simple

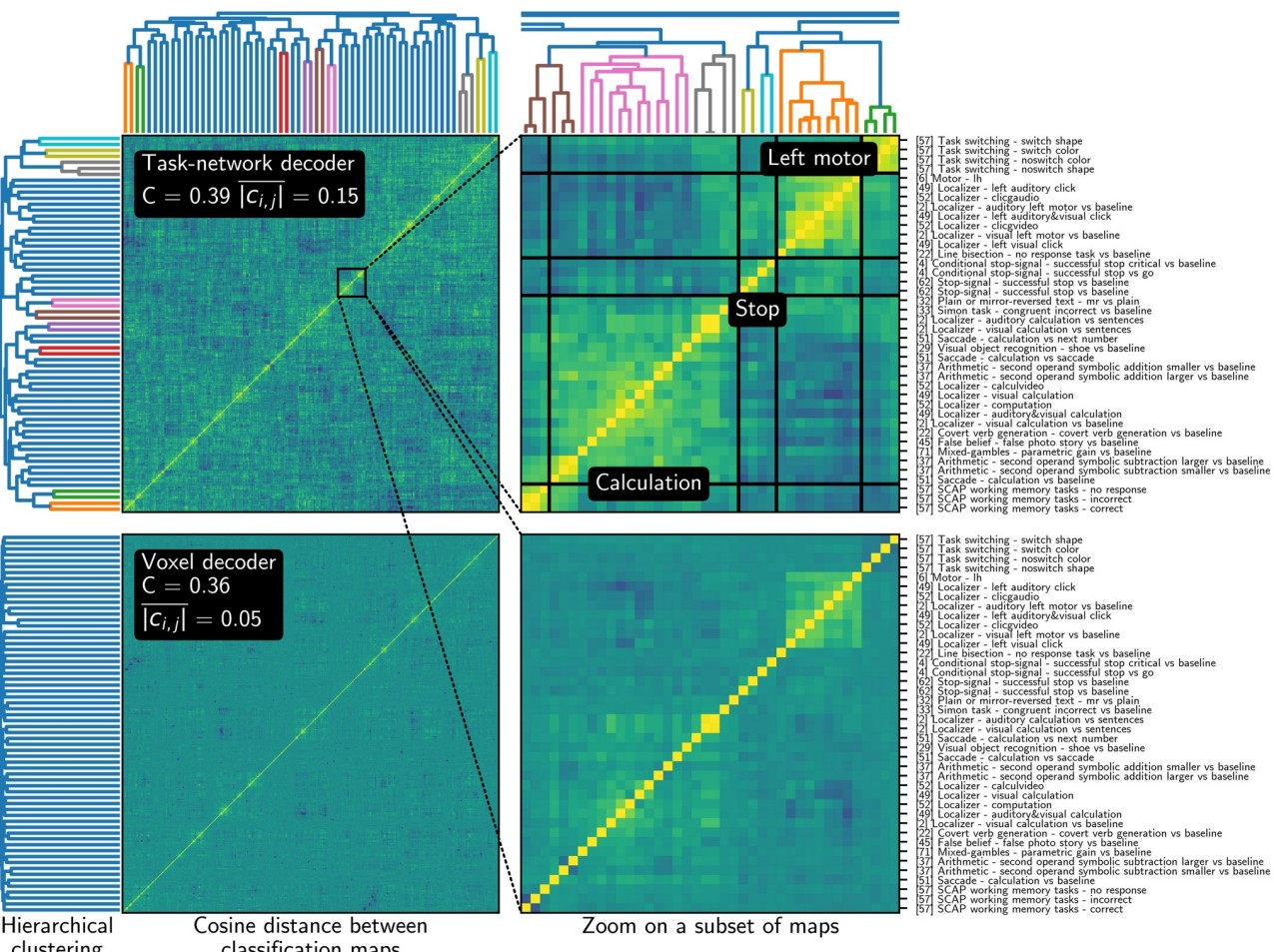

**Fig 7. Cosine similarities between classification maps, obtained with our multi-study decoder (top) and with decoders learned separately (bottom), clustered using average-linkage hierarchical clustering.** The classification maps obtained when decoding from task-optimized networks are more easily clustered into cognitive-meaningful groups using hierarchical clustering—the cophenetic coefficient of the top clustering is thus higher. Maps may also be compared using the similarities of their loadings on MSTONs, with similar results.

functional networks decoders. Classification maps learned using task-optimized networks are more correlated on average, and hierarchical clustering reveals a sharper correlation structure. This is because the whole classification matrix is low-rank (rank $l = 128 < c = 545$) and influenced by the many studies we consider—the classification maps of our model are supported by networks relevant for cognition. As a consequence, it is easier to cluster maps into meaningful groups using hierarchical clustering based on cosine distances. For instance, we outline inter-study groups of maps related to left-motor functions, or calculation tasks. Hierarchical clustering on baseline maps is less successful: the associated dendrogram is less structured, and the distortion introduced by clusters is higher (as suggested by the smaller cophenetic coefficient). Clusters are harder to identify, due to smaller contrast in the correlation matrix. Multi-study training thus acts as a regularizer, by forcing correlation across maps with discovered relations. This regularization partly explains the increase in decoding accuracy.

## Discussion

The methodology presented in this work harnesses the power of deep representations to build multi-study decoding models for brain functional images. It brings an immediate benefit to

functional brain imaging by providing a universal way to improve the accuracy of decoding in a newly acquired dataset. Decoding is a central tool to draw inferences on which brain structures implement the neural support of the observed behavior. It is most often applied to task-fMRI studies with 30 or less subjects, which tend to lack statistical power [72]. In this regime, aggregating existing studies to a new one using a multi-study model as the one we propose is likely to improve decoding performance. This is further evidenced in Fig B in S1 Appendix: using MSTONs as a decoding basis on a new decoding task outperforms using resting-state networks. Of course, such improvement can only occur if the cognitive functions probed by the new study are related to the ones probed in the multi-study corpus. We foresee limited benefits when analyzing strongly original task fMRI experiments, and experiments studying very specific and high-level cognitive functions, that MSTONs are only partially able to capture (Fig 5).

With increasing availability of shared and normalized data, multi-study modeling is an important improvement over simple decoders, provided that it can adapt to the diversity of cognitive paradigms. Our *transfer-learning* model has such flexibility, as it does not require explicit correspondence across experiments. Beyond quantitative benefits –the gain in prediction accuracy– the models also brings qualitative benefits, facilitating the interpretation of decoding maps (Fig 6). Pooling subjects across studies effectively increases the sample size, as advocated by [2]. The resulting increase in statistical power for cognitive modeling will help addressing the reproducibility challenge outlined by [3]. In our setting, each study (or *site*) provides a single decoding objective, which is predicting one contrast among all other contrasts from this study. This is a validated approach in decoding [73]. As some studies use different fMRI tasks, we may also use one decoding objective per *task*, with similar quantitative improvement in performance (see Fig F in S1 Appendix).

Our modeling choices were driven by the recent successes of deep non-linear models in computer vision and medical imaging. However, we were not able to increase performance by departing from linear models: introducing non linearities in our models provides no improvement on left-out accuracy. On the other hand, we have shown that pooling many fMRI data sources enables to learn deeper models, although these remain linear. Techniques developed in deep learning prove useful to fit models that generalize well across subjects: using dropout regularization [14] and advanced stochastic gradient techniques [13] is crucial for successful transfer and good generalization performance.

Sticking to linear models brings the benefit of easy interpretation of decoding models. The use of sparsity and non-negativity in the training and consensus phase allow to obtain interpretable networks. Using sparsity only in each phase (as originally advocated by [74]) yields "contrast" networks with both positive and negative regions, that are harder to interpret (see also [60]). In particular, this limits the occurence of non-zero weights that reflect noise suppression [75].

The models capture information relevant for many decoding tasks in their internal representations. From these internals, we extract interpretable cognitive networks, inspired by matrix factorization techniques used to interpret computer vision models [76]. The good predictive performance of MSTONs networks (Fig 2 and Fig B in S1 Appendix) provides quantitative support for their decomposition of brain function. Extracting a universal basis of cognition is beyond the scope of a single fMRI study, and should be done by analysis across many studies. We show that, across studies, a joint predictive model finds meaningful approximations of atomic cognitive functions spanning a wide variety of mental processes (Fig 4). This methodology provides a step forward towards defining mental processes in a quantitative manner, which remains a fundamental challenge in psychology [9, 77]. Yet, in the present work, the delineation of atomic cognitive functions remains coarse and incomplete. This is

likely due to the limited scope of our corpus, and to the fact that we automatically align the cognitive functions probed by the various studies of the corpus. Expert annotation of mental process involved in the studies could greatly help establishing a clearer picture.

Our approach differs from commonly-used decomposition techniques in fMRI analysis (e.g. ICA [78], or dictionary learning [74]), that are used to extract *functional networks*. These techniques optimize an unsupervised reconstruction objective over resting-state data, in effect capturing co-occurrence of brain activity across distributed locations. They have traditionally been used with few components (e.g. $k \approx 20$). In contrast, after the first decomposition, performed without information from the tasks, we extract the MSTONs components to optimize the decoding performance on many tasks. Leaving a systematic comparison between MSTONs and classical functional networks for future work, we already make two observations. First, a fraction of functional networks extracted by unsupervised methods support non-Gaussian noise patterns in the BOLD time-series, and permits noise suppression [79, 80]. Typically, only a fraction of the networks extracted in an ICA analysis is interpreted. MSTONs, on the other hand, optimize a supervised objective and focus on the fraction of the BOLD signal related to the tasks. Second, MSTONs (despite being more noisy) appears more skewed towards known coordinated brain networks (Figs 4 and 5), that differs from the networks recruited at rest (see e.g. [81] for a comparison of task and rest brain networks).

We use many different fMRI studies to distill MSTONs across various tasks. This data aggregation approach requires little supervison. The flip side is that it leads to coarse results by nature: our approach is obviously not sufficient to recover the detailed brain-to-mind mapping, collective knowledge of psychologists and neuroscientists, that has emerged from decades of research on multimodal datasets and careful behavioral experiments. Specific brain-to-mind associations are best resolved with dedicated experiments using experimental-pyschology paradigms tailored to the question at hand. Other data than fMRI, for instance more invasive, may also provide stronger evidence. For instance a double dissociation in brain-lesion patients give unambiguous evidence of distinct cognitive processes via distant neural supports, as with Broca and Wernicke's separation of language understanding and generation [82], or the more recent teasing out of emotional and cognitive empathy [83].

Finally, the current version of our framework does not model explicit inter-subject variability, and is rather focused on extracting commonalities across subjects. Future work may augment multi-study decoding with such information, as obtained by e.g., hyperalignment techniques [84].

## Conclusion

The success of using distributed representations to bridge cognitive tasks supports a system-level view on how brain activity supports cognition.

Our multi-study model will become increasingly useful to brain imaging as the number of available studies grows. Such a growth is driven by the steady increase of publicly shared brain-imaging data, facilitated by online neuroimaging platforms and increased standardization [2, 85]. With a larger corpus of studies, the proposed methodology has the potential to build even better universal priors that overall improve statistical power for functional brain imaging. As such, multi-study decoding provides a path towards knowledge consolidation in functional neuroimaging and cognitive neuroscience.

## Supporting information

**S1 Appendix. Detailed methods.** This appendix discusses technical details of the multi-study decoding approach: the specific architecture, a 3-layer linear model, and the deep-learning

technique used to regularize and train it. **Discussion on the model design**. In this appendix, we perform supportive experiments to explain the observed results, An ablation study of the various model components is provided to further support modelling choices. **Reproduction details and tables**. In this appendix, we provide implementation details for reproducibility, along with tables with quantitative results per contrast.
(PDF)

**S1 Components. This file holds a visualization of all the multi-study task optimized networks that we introduce in this paper.**
(ZIP)

## Author Contributions

**Conceptualization:** Arthur Mensch, Julien Mairal, Bertrand Thirion, Gaël Varoquaux.

**Data curation:** Arthur Mensch.

**Formal analysis:** Arthur Mensch, Julien Mairal, Bertrand Thirion, Gaël Varoquaux.

**Funding acquisition:** Bertrand Thirion, Gaël Varoquaux.

**Investigation:** Arthur Mensch.

**Methodology:** Arthur Mensch, Julien Mairal, Bertrand Thirion, Gaël Varoquaux.

**Project administration:** Bertrand Thirion, Gaël Varoquaux.

**Resources:** Arthur Mensch, Bertrand Thirion, Gaël Varoquaux.

**Software:** Arthur Mensch, Bertrand Thirion, Gaël Varoquaux.

**Supervision:** Julien Mairal, Bertrand Thirion, Gaël Varoquaux.

**Validation:** Arthur Mensch.

**Visualization:** Arthur Mensch, Gaël Varoquaux.

**Writing – original draft:** Arthur Mensch.

**Writing – review & editing:** Arthur Mensch, Julien Mairal, Bertrand Thirion, Gaël Varoquaux.

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
