## [Decision Letter · Decision Letter 0]

10 Jun 2020

Dear Dr Mensch,

Thank you very much for submitting your manuscript "Extracting Representations of Cognition across NeuroImaging Studies Improves Brain Mapping" for consideration at PLOS Computational Biology.

Sorry for the delay in this first decision. One of the reviewers became unresponsive. After several unanswered reminders (and after being sure they were alive and well, given their activity on other platforms) we decided to remove them. Of course I take responsibility for this delay, and please accept my apologies.

As with all papers reviewed by the journal, your manuscript was reviewed by members of the editorial board and by several independent reviewers. A general appreciation for an ambitious study is paired to several major concerns, both on the method and the presentation. In light of the reviews (below this email), we would like to invite the resubmission of a significantly-revised version that takes into account the reviewers' comments.

We cannot make any decision about publication until we have seen the revised manuscript and your response to the reviewers' comments. Your revised manuscript is also likely to be sent to reviewers for further evaluation.

Sincerely,

Daniele Marinazzo

Deputy Editor

PLOS Computational Biology

Reviewer's Responses to Questions

**Comments to the Authors:**

Reviewer #1: Mensch et al. present an approach that leverages transfer learning (an important and relatively recent topic within machine learning) for functional brain imaging with the aim to extract shared information across many different experimental paradigms. Their deep linear model (i.e., three layers) is applied to 35 different fMRI studies spanning 545 different experimental contrasts. The method results in 128 functional task-related brain networks that can be interpreted by themselves but importantly, also can be reused in new data analyses with the idea to improve decoding accuracy especially for studies suffering from low sample sizes.

This is a very well-written and structured manuscript of general interest for the neuroimaging community. The method definitely addresses an important issue for cognitive neuroscience research: extracting shared and interpretable representations across cognitive paradigms in a purely data-driven manner. There is a great consensus in the field that understanding the functional role of brain networks requires understanding their involvement over a large space of cognition. Their code and results are available on github and the resulting brain networks are visualized on a dedicated website and can also be downloaded from there.

However, I have a few points that need to be addressed before I can recommend publication.

1. The choice of the dimensions in the first (i.e., 512 resting-state networks) as well of second layer (i.e., 128 task networks) in their model seems arbitrary.

a. How robust are resulting task-optimized networks for different decisions for k (i.e., 512) and l (i.e., 128)?

b. My main criticism is that the large number of resulting task networks makes interpretation of the networks actually not straightforward. What would be the effect of lowering l? Also, when inspecting all networks on their website, I noticed that ¬many components feature the same contrast as being the one that increases its likelihood the most (e.g. “house” for components 3/22/48/54/62/67 or “saccades” for components 43/68/78/124). Please discuss this.

c. What would be the effect of enforcing more/less overlap between the networks?

d. Please discuss if resulting networks can be interpreted straightforward or if they have a similar problem as weight maps for LDA/SVM have; where non-zero weights could also reflect noise suppression in that brain region (Haufe et al. 2014).

2. With regards to methods, I feel some parts need some clarification/additional information.

a. Each study is treated as a classification task and this implies that each study differs in the number of contrasts that needs to be classified; i.e., differs in its difficulty for decoding and its respective chance level. I would suggest making this more explicit in the manuscript (it was not immediately clear to me) and to highlight this in Fig. 2A by aligning studies according to their chance level and not absolute decoding accuracy.

b. Further with respect to my point in 2a, it was not clear to me if the classification objective in equation (4) (p.21) somehow takes into account the number of contrast that needs to be decoded on the study-level and does not skew optimization towards improving “easier” studies with fewer contrasts to decode (e.g., rhyme judgement with only 3 contrasts)?

c. It would be also informative to see the decoding accuracy for contrasts separately and not just study-level and cluster them in a meaningful way (contrast loading on the same task-optimized network?). This could provide important insights into which contrasts are harder/easier to decode in the context of multi-study decoding.

3. In the Discussion section authors could elaborate on a few more points

a. Why do the authors think that contrasts that are moderately difficult to decode benefit from multi-study decoding? Is there a link with the number of subjects for those kind of contrasts (i.e., are moderate difficult to decode contrasts more prominent for studies with small N)?

b. What is the effect of non-negative matrix factorization? Is this a valid assumption?

Minor:

1. The list of contrast per study and the number of actual contrasts investigated (i.e., Supplementary Table 1) should be moved to a more prominent part in the manuscript.

2. That networks can be downloaded and visually inspected on their website should be mentioned in the Abstract and manuscript.

3. Ensemble model: In the results, this is referred to as consensus model and I would advise to mention this terminology in the main method section to avoid confusion.

4. Please mention how it was determined how many subjects were left out per study given difference in N (p.19)?

5. The authors should consider moving important methodological details (e.g., the three-layer model description, on dropout and low rank constraints) to the main method section.

6. The figures need clarification:

a. Fig. 3: Does standard decoder refer to voxel-level or resting-state? Please label more precisely.

b. Fig. 4: Word clouds contain labels of contrasts, e.g. C01, C02 that are meaningless for a reader.

c. Fig. 5: What do blue and red colors refer to (positive and negative weights?)? How can the accuracy gain be +6.9% when baseline accuracy is 99.4% for Face vs House?

7. Identified a few spelling mistakes:

a. P.20, line 657 “on on”

b. P.13, line 381: remove dot

c. P.33, line 936: “T he”

Reviewer #2: see attached

Reviewer #3: In this manuscript, Mensch and colleagues seek to determine a machine learning approach that simultaneously resolves significant limitations in the field of cognitive neuroscience: the Many-to-one mappings of function to structure and the few cognitive tasks administered in a single experiment. Limitations on sample sizes further challenge attempts for an integrative approach to determine the brain’s functional neuroanatomy using machine learning. The authors use a multi-task linear decoding model, which integrates resting-state functional connectivity networks, to determine reliable patterns of functional activity driven by cognition as a tool for decoding subsequent studies.

The authors have set up a very ambitious agenda, to determine “universal cognitive representations” and suggest “While extracting a universal basis of cognition is beyond the scope of a single fMRI study, we show that a joint predictive model across multiple studies finds meaningful approximations of atomic cognitive function”. However, careful examination of Figure 4 suggests a starkly different reality. Linking motor cortex to motor behaviour is good proof of concept, but is not a cognitive neuroscience discovery. Outside of motor behaviour, other cognitive functions not meaningfully represented. What stands out is how little meaningful information can be extracted from the word clouds aside from rudimentary functional localization, for example faces and fusiform cortex, with conceptual noise (associated labels).

My primary concern involves the task contrasts that are entered into the model and the use of the cognitive labels. Are the task contrasts used in the current study following the experimental intent of using a tight contrast to determine precise cognitive engagement of brain regions? Or, are the authors contrasting every condition with every other condition for each subject? For example, in the HCP data, social cognition can be “localized” by comparing the mental interaction blocks with the random interaction blocks. However, contrasting geometric shapes moving with intention with tools from the working memory condition will not provide meaningful neural information about mentalizing or tools. This would confound the labels and set up a “garbage in” scenario. I do not know if this contrast was used to train the model. 23 contrasts were used from HCP but no additional detail is provided in supplemental. The tasks of interest in the appendix are listed by site or task. However, many sites have multiple tasks. All tasks, as well as all contrasts, should be listed.

There is little information linking executive function or memory to the brain. These higher order cognitive functions have reliable patterns of brain activity. Conversely, cognitive functions linked to the frontoparietal control or default mode networks would be interesting due to their positioning within the cortical hierarchy. Is there some reason they are not depicted? Does this point to a limitation of the method?

The authors suggest that fMRI studies of 30 subjects or less “lack statistical power”. It is more accurate to say they have low power. The authors additionally suggest in the introduction that power has been decreasing in cognitive neuroscience. This is not exactly accurate. The effect sizes that cognitive neuroscientists are investigating are smaller, which is true for most maturing fields of science. So, while the effect sizes are smaller, sample size is most certainly increasing. It is possible that the ratio of smaller effect sizes to increases in sample is overall decreasing power. However, this is a matter of debate and likely varies across areas of inquiry. More thoughtful nuance would be appreciated. The authors’ general tone toward the field is somewhat disrespectful, which is surprising considering how reliant the authors are on these data.

It is unclear how much of the current work is an innovation, or an incremental improvement on the tools and algorithms developed over many years by the research team. This should be spelled out much more clearly in the introduction and discussion.

**Have all data underlying the figures and results presented in the manuscript been provided?**

Reviewer #1: Yes

Reviewer #2: Yes

Reviewer #3: Yes

PLOS authors have the option to publish the peer review history of their article (what does this mean?). If published, this will include your full peer review and any attached files.

Reviewer #1: Yes: Romy Lorenz

Reviewer #2: Yes: Savannah L. Cookson

Reviewer #3: No
---

## [Decision Letter · Decision Letter 1]

4 Oct 2020

Dear Dr Mensch,

Thank you very much for submitting your manuscript "Extracting representations of cognition across neuroimaging studies improves brain decoding" for consideration at PLOS Computational Biology.

While we and the reviewers appreciated the revisions, and most of the issues have been clarified, one major comment pointing to a possibly major limitation of your work still needs to be properly addressed.

We cannot make any decision about publication until we have seen the revised manuscript and your response to the reviewers' comments. Your revised manuscript is also likely to be sent to reviewers for further evaluation.

Sincerely,

Daniele Marinazzo

Deputy Editor

PLOS Computational Biology

Daniele Marinazzo

Deputy Editor

PLOS Computational Biology

Reviewer's Responses to Questions

**Comments to the Authors:**

Reviewer #1: The authors responded to my comments satisfactorily.

Reviewer #2: The authors have done an exemplary job responding to my and the other reviewers' critiques. I have no further comments.

Reviewer #3: In the response to reviewers, the authors omitted my previous comment: “There is little information linking executive function or memory to the brain. These higher order cognitive functions have reliable patterns of brain activity. Conversely, cognitive functions linked to the frontoparietal control or default mode networks would be interesting due to their positioning within the cortical hierarchy. Is there some reason they are not depicted? Does this point to a limitation of the method?”

There does seem to be a partial reply appended beneath the previous one stating: “Fig. 4 only shows a selection of networks. The orbitoparietal networks and default mode networks also appears among the MSTONs (see https://cogspaces.github.io/assets/MSTON/components.html, for instance components 10 and 24), but are associated with many different contrast names. We therefore deamed them less interesting to be shown in the main text. We have insisted on this point in the text and added a supplementary figure (Fig. 7) with the MSTONs cited above, with this discussion (Appendix S2.1.1). “

However, this statement is not responsive. On examination of Supplemental Figure 7, it is stated “Fig. 7. Examples of MSTONs that recruits well-known patterns of brain activity.: A. Default Mode Networks and B. Right orbitofrontal cortex” However, the maps are inconsistent with these anatomical terms, and the corresponding word clouds inconsistent with the functions of the named networks. A. Default Mode Networks depicts middle cingulate with the executive function “Switch and dual task probe” (NOTE: Core regions of the default network are posterior cingulate and medial prefrontal cortex and the inferior parietal lobule; Buckner et al., 2008, ANYAS), while the Right orbitofrontal cortex image depicts not orbitofrontal cortex but a subcortical region with the term “Bottle”. Were these figures generated in error? Or, are the authors asserting that these “well-known patterns of brain activity” are consistent with the cognitive neuroscience literature? If the latter is the case, then this needs to be supported by references to the literature. The structure function associations suggested in this figure are not well-known to this reviewer.

On page 9, the authors write, “Some MSTONs are less specific to particular contrasts, but correspond to reliable patterns of brain activity during task: for example, the default mode networks and the frontoparietal cortex appear saliently (App. S2.1.1, Fig 7).”

Again, Figure 7 is not what the authors suggest they are depicting, and there is no depiction of the frontoparietal cortex at all. I would like to see the default mode network and the frontoparietal control networks’ topography and functions accurately depicted.

In appendix 2.1.1, the author write: “Well-known patterns of brain activity. Some MSTONs are also composed of brain regions that are activated in many different tasks, such as the default mode networks or the orbitofrontal cortex. We show two examples of such networks in Fig 7, along with word-clouds of associated contrasts. As expected, the associated contrasts do not correspond to identifiable brain functions.”

There appears to be a significant contradiction between the notion that there are “Examples of MSTONs that recruits well-known patterns of brain activity” AND “As expected, the associated contrasts do not correspond to identifiable brain functions.”

I return to my original critique…The method employed here draws in information linking executive function or memory to the brain. These higher order cognitive functions have robust and reliable patterns of brain activity. Conversely, cognitive functions linked to the frontoparietal control or default mode networks are also reliable and robust. Convergence between these cognitive functions and brain regions have been demonstrated from a century of lesion work, 3 decades of neuroimaging, and more recently with intracranial EEG. If the present approach is unable to recapitulate them, there is a problem with the method. These cognitive functions and networks should be depicted and in a transparent manner (not spread between the main text, appendices, and supplemental figures). If the method cannot associate the two, this is a very significant limitation that needs to be openly acknowledged and discussed in the main text.

**Have all data underlying the figures and results presented in the manuscript been provided?**

Reviewer #1: Yes

Reviewer #2: Yes

Reviewer #3: None

PLOS authors have the option to publish the peer review history of their article (what does this mean?). If published, this will include your full peer review and any attached files.

Reviewer #1: **Yes: **Romy Lorenz

Reviewer #2: **Yes: **Savannah L. Cookson

Reviewer #3: No
---

## [Decision Letter · Decision Letter 2]

25 Nov 2020

Dear Dr Mensch,

Thank you very much for submitting a revised version of your manuscript "Extracting representations of cognition across neuroimaging studies improves brain decoding" for consideration at PLOS Computational Biology.

Unfortunately I have to get back to you with another revision request.

There are some significant discrepancies between the response letter, where several limitations are acknowledged, and the main text, where the same are discussed only marginally if at all.

This is problematic for two main reasons: first of all this is not a correct scholarly and scientific practice, which also could put in a different perspective the errors and omissions in the previous rounds of review.

Secondly, this is about the scope and significance of your work. Your methods is elegant, correct, and also reasonably satisfactory in some applications, when trained on a significant yet non-universal number of tasks. You should appreciate the fact that colleagues and readers more familiar with a different way of approaching neuroscience research might wonder how universal your approach is, whether it works better in some cases than others, and why. We think that the comment of the reviewer on "decades of research with multimodal datasets" refers to the collective knowledge of psychologists and neuroscientists, which you should consider as a perspective, and not as a benchmark against which you would be requested to compete, we agree that this is not the case.

We wish to give you a last opportunity to be more critical with your approach, confident that not just the scientific process, but also the impact of your paper on a wider audience, will be benefited.

Sincerely,

Daniele Marinazzo

Deputy Editor

PLOS Computational Biology

Daniele Marinazzo

Deputy Editor

PLOS Computational Biology

Reviewer's Responses to Questions

**Comments to the Authors:**

Reviewer #3: The authors have been somewhat responsive to the prior round of review, particularly if re-review was restricted to the response letter. I appreciate the acknowledgement and correction of errors identified in the prior submission. The revised figure 5 provides a more substantive response to the issue of functional correspondence of 1) higher order cognitive processes and 2) distributed/heteromodal brain networks (default, frontoparietal).

In my prior review I stated “If the method cannot associate the two, this is a very significant limitation that needs to be openly acknowledged and discussed in the main text.”

In the response to reviewers letter, the authors clearly state this limitation:

“As we now clearly acknowledge in the main text, our interpretation effort is more successful for lower-level than for higher-level brain functions.”

“As discussed above, some MSTONs corresponding to low-level cognitive functions, are related to easily interpretable set of tasks. MSTONs corresponding to higher-level functions are related to a more diffuse set of tasks, harder to interpret.”

“We now acknowledge in the main text (around line 331) that our word-cloud generation approach provides more interesting results for lower-level than for higher level MSTONs.”

However, in the manuscript, the authors DO NOT convey this limitation as candidly. The only mention of the limitation “around line 331” is in the results, not discussion. The authors state:

“ The MSTONs each plays a role in decoding a subset of 323 contrasts. They capture both low-level and high-level cognitive signal.”

“The components related to low-level functions tend to have structure-function links that are easier to interpret, high-level functions are also well captured.”

There is no revision to the discussion, which was requested by this reviewer. Instead, the authors make two sweeping claims in the discussion without caveat or clarification:

1) “While extracting a universal basis of cognition is beyond the scope of a single fMRI study, we show that a joint predictive model across multiple studies finds meaningful approximations of atomic cognitive functions.”

To reiterate, the authors assert they have approximated the atoms of cognitive function. Yet in the response to reviewers, the authors write “However, we find that the reviewer’s expectation do not do justice to the work: a single data processing can hardly be faithful to a history of investigations across multiple data modalities.” The data modalities the authors refer to are converging lines of evidence linking structure to function, including fMRI, and fMRI tasks that are included in the current work.

2) “In this regime, aggregating existing studies to a new one using a multi-study model as the one we propose is likely to improve decoding performance.”

But what if the new study involves memory, executive control, or some other construct not well-captured by the MSTONs? What is the utility of this tool, particularly with novel cognitive neuroscience investigations that are poorly captured by MSTONs?

Also in the discussion, the authors write: “Second, MSTONs appears more skewed towards known coordinated brain networks (Fig 4 and Fig 5), that differs from the networks recruited at rest.”

This statement is factually inaccurate. First, there is a not more noise in the spatial distribution of the MSTONs, which renders the phrase “known coordinated brain networks” untrue. Second, resting-state networks have been robustly related to task-based coactivation patterns (Smith et al 2009 PNAS, https://pubmed.ncbi.nlm.nih.gov/19620724/; Laird et al 2011 JOCN, https://pubmed.ncbi.nlm.nih.gov/21671731/).

When the authors wrote in their response letter, “We have addressed the substance of the reviewer’s comments by modifying the manuscript in multiple places, including reworking completely our discussion on the high-level functional structures present in the MSTONs (fig 5), and better discussions on the interpretations of our results.” I expected revision to the discussion section of the manuscript. This did not happen. I fully agree with the authors that “it is non a trivial task to start from functional-imaging data without prior information and decompose the brain in networks that predict the mental processes underway, from low-level to high-level.” However, the authors claims in the discussion are too broad, and the potential limitations are not sufficiently acknowledged.

**Have all data underlying the figures and results presented in the manuscript been provided?**

Reviewer #3: Yes

PLOS authors have the option to publish the peer review history of their article (what does this mean?). If published, this will include your full peer review and any attached files.

Reviewer #3: No
---

## [Decision Letter · Decision Letter 3]

15 Feb 2021

Dear Dr Mensch,

After an editorial consultation, we decided that your manuscript 'Extracting representations of cognition across neuroimaging studies improves brain decoding' is provisionally accepted for publication in PLOS Computational Biology.

We took this decision in the light of the amount of work done, the effort towards improvement in a difficult field, and the committment towards an open approach. We hope that the improvement can continue from here.

Best regards,

Daniele Marinazzo

Deputy Editor

PLOS Computational Biology

Daniele Marinazzo

Deputy Editor

PLOS Computational Biology

Reviewer's Responses to Questions

**Comments to the Authors:**

Reviewer #3: The authors appear to have contextualized their findings, as well as acknowledge that there may be some limitations to the work. In this revision, there is some movement on the part of the authors in noting challenges, particularly regarding the MSTONs corresponding to higher-level functions. The authors note in the discussion that some expert annotation would “greatly help establishing a clearer picture (line 453)”. I agree. Moving beyond motor and perceptually driven behaviors, the authors push forward a lot of garbage terms that are clearly not psychological processes (e.g. “explode”, “C01”, “C02”, “C03”, “junk”, “distance from indifference”). Of the approximately 50 terms associated with three default mode network components, only two are consistent with meta-analytic results derived from Neurosynth. It is worth noting, however, that for one default mode network component, the experimental condition (False belief) appears along with this task’s comparison condition (False photo), and I do not understand what this is supposed to mean. The authors purport in the abstract that “The extracted networks have been made available; they can be readily reused in new neuro-imaging studies. We provide a multi-study decoding tool to adapt to new data”. However, I find that the significant limitations of this tool will render it of little utility to cognitive neuroscientists.

**Have all data underlying the figures and results presented in the manuscript been provided?**

Reviewer #3: Yes

PLOS authors have the option to publish the peer review history of their article (what does this mean?). If published, this will include your full peer review and any attached files.

Reviewer #3: No

---

## [Editor Report · Acceptance letter]

7 Apr 2021

PCOMPBIOL-D-20-00479R3 

Extracting representations of cognition across neuroimaging studies improves brain decoding

Dear Dr Mensch,

I am pleased to inform you that your manuscript has been formally accepted for publication in PLOS Computational Biology. Your manuscript is now with our production department and you will be notified of the publication date in due course.

With kind regards,

Alice Ellingham
